# Depressive symptoms and their association with age, chronic conditions and health status among middle-aged and elderly people in peri-urban Tanzania

comorbidities of depression; depression; older adults; Tanzania; sub-Saharan Africa

**Corresponding author:**
Stefan Kohler;
Email: stefan.kohler@uni-heidelberg.de

Laura-Marie Stieglitz[1] , Leslie B. Adams[2] , Till Bärnighausen[1] , Anne Berghöfer[3] , Patrick Kazonda[4] , Japhet Killewo[5] , Germana H. Leyna[5] , Julia Lohmann[1,6] , Julia K. Rohr[7]  and Stefan Kohler[1,3] 

[1]Heidelberg Institute of Global Health, Faculty of Medicine and University Hospital, Heidelberg University, Heidelberg, Germany; [2]Department of Mental Health, Johns Hopkins Bloomberg School of Public Health, Baltimore, MD, USA; [3]Institute of Social Medicine, Epidemiology and Health Economics, Charité – Universitätsmedizin Berlin, Berlin, Germany; [4]Dar es Salaam Urban Cohort Study, Dar es Salaam, Tanzania; [5]Department of Epidemiology and Biostatistics, Muhimbili University of Health and Allied Sciences, Dar es Salaam, Tanzania; [6]Department of Global Health and Development, London School of Hygiene & Tropical Medicine, London, UK and [7]Harvard Center for Population and Development Studies, Harvard University, Cambridge, MA, USA

## Abstract

**Background:** Depression is a global mental health challenge. We assessed the prevalence of depressive symptoms and their association with age, chronic conditions, and health status among middle-aged and elderly people in peri-urban Dar es Salaam, Tanzania.
**Methods:** Depressive symptoms were measured in 2,220 adults aged over 40 years from two wards of Dar es Salaam using the ten-item version of the Center of Epidemiologic Studies Depression Scale (CES-D-10) and a cut-off score of 10 or higher. The associations of depressive symptoms with age, 13 common chronic conditions, multimorbidity, self-rated health and any limitation in six activities of daily living were examined in univariable and multivariable logistic regressions.
**Results:** The estimated prevalence of depressive symptoms was 30.7% (95% CI 28.5–32.9). In univariable regressions, belonging to age groups 45–49 years (OR 1.35 [95% CI 1.04–1.75]) and over 70 years (OR 2.35 [95% CI 1.66–3.33]), chronic conditions, including ischemic heart disease (OR 3.43 [95% CI 2.64–4.46]), tuberculosis (OR 2.42 [95% CI 1.64–3.57]), signs of cognitive problems (OR 1.90 [95% CI 1.35–2.67]), stroke (OR 1.56 [95% CI 1.05–2.32]) and anemia (OR 1.32 [95% CI 1.01–1.71]) and limitations in activities of daily living (OR 1.35 [95% CI 1.07–1.70]) increased the odds of depressive symptoms. Reporting good or very good health was associated with lower odds of depressive symptoms (OR 0.48 [95% CI 0.35–0.66]). Ischemic heart disease and tuberculosis remained independent predictors of depressive symptoms in multivariable regressions.
**Conclusion:** Depressive symptoms affected almost one in three people aged over 40 years. Their prevalence differed across age groups and was moderated by chronic conditions, health status and socioeconomic factors.

## Impact statement

Depression and aging are global public health concerns that can be intertwined. In later life, depression can threaten physical, cognitive and social functioning and lead to higher mortality and healthcare costs. Older adults with chronic conditions are at a higher risk for depression, and the co-occurrence of chronic conditions and depression can worsen the course and outcome of both. As more people reach higher ages in sub-Saharan Africa, health systems face the need to expand services for elderly people who live with depression and chronic health conditions. This study of depressive symptoms and their relation to age, chronic diseases and health status reports data from a community-based sample of middle-aged and elderly adults living in two peri-urban wards of Dar es Salaam, Tanzania. The presented findings can help identify people at risk for depression and inform health system planning.

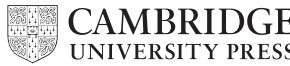

## Introduction

In the Global Burden of Disease Study 2019, 5.7% of those aged over 60 years were estimated to be affected by depression worldwide. The prevalence of depression was estimated to be highest in sub-Saharan Africa, where depression might have affected 8.4% of the elderly

before the COVID-19 pandemic (GBD Mental Disorders Collaborators, 2022). Depression has been associated with decreased physical, cognitive and social functioning and increased mortality in elderly people (Sinclair et al., 2001; Schillerstrom et al., 2008; Heward et al., 2018; Wei et al., 2019). Higher health services utilization and healthcare costs have further been reported for adults with depression (Luppa et al., 2012; Konig et al., 2019). To build capacity in health systems for the treatment and prevention of depression, context-specific knowledge about the prevalence of depressive symptoms and their association with age, chronic conditions and the current health status is useful.

Depression has been linked to chronic physical conditions like obesity (de Wit et al., 2010), diabetes (Renn et al., 2011), hypertension (Meng et al., 2012), stroke, cardiac disease, lung disease (Huang et al., 2010) and cognitive impairment (Pellegrino et al., 2013). HIV and tuberculosis have also been associated with an increased risk and worsened outcome of depression (Antelman et al., 2007; Sweetland et al., 2017). Chronic conditions can affect a person's subjective health status and might be the underlying causes why, for instance, poor self-rated health and limitations in activities of daily living have been associated with an increased risk for depression among the elderly (Cole and Dendukuri, 2003; Chang-Quan et al., 2009; Kwon et al., 2019). Studies of urbanization and depression suggest that their relationship differs across geographic regions and income levels (Penkalla and Kohler, 2014; Sampson et al., 2020). While depression and aging are global public health concerns, we know of only few previous studies that assessed the relationship of depression with chronic conditions in middle-aged and elderly populations in sub-Saharan Africa.

Two community-based studies examined depression among the elderly in rural areas of Tanzania. One study estimated the prevalence of depression at 44.4% among those aged over 60 years based on the 15-item Geriatric Depression Scale (Adams et al., 2020). The other study reported a prevalence of 21.2% among those aged over 70 years based on a clinical assessment of depression (Mlaki et al., 2021). We estimated that depressive symptoms affected 32.5% of ≥40-year-olds in a study examining comorbidities among women in peri-urban Dar es Salaam. Depressive symptoms affected 56.4% of women with ischemic heart disease, 55.0% of women with tuberculosis and 46.7% of women with signs of cognitive problems (Stieglitz et al., 2022). Chronic conditions that have been associated with depression in other studies from sub-Saharan Africa include angina, heart disease, asthma, arthritis, sleep problems and stroke (Peltzer and Phaswana-Mafuya, 2013; Ojagbemi et al., 2017; Brinkmann et al., 2020). Studies from rural Burkina Faso and rural South Africa have found no evidence of association of depressive symptoms with HIV, obesity, hypertension, diabetes and the metabolic syndrome (Geldsetzer et al., 2019; Brinkmann et al., 2020).

In this study, we assessed depressive symptoms among middle-aged and elderly women and men who live in a peri-urban area of Tanzania. First we estimated the prevalence of depressive symptoms in ≥40-year-olds in the Ukonga and Gongolamboto wards in Dar es Salaam. Second we examined the associations of depressive symptoms with age, 13 chronic conditions, multimorbidity, self-rated health and limitations in activities of daily living.

## Methods

### *Study design and setting*

A cross-sectional study was conducted in the neighboring Ukonga and Gongolamboto wards of Dar es Salaam, the largest city in Tanzania, between June 2017 and July 2018. Both wards were densely populated and located about 20 km from the city center. The wards comprised a mixture of well-built houses and mud houses. Almost all households had a toilet and about half had electricity supply (Leyna et al., 2017). The study was part of the 'Health and Aging in Africa: Longitudinal Studies in three INDEPTH Communities' (HAALSI) project with further research sites in South Africa and Ghana. It was nested in the Dar es Salaam Health and Demographic Surveillance System, which is also known as the Dar es Salaam Urban Cohort Study (DUCS).

### *Study population and sample selection*

A sex-stratified, random sample of 2,450 men and 2,400 women aged over 40 years in 2017 was selected from the DUCS 2016 population. Half of this sample was randomly selected for point-of-care blood tests. Of the 4,850 identified study participants, 2,299 (744 men and 1,555 women) could be surveyed and 1,024 (318 men and 706 women) recruited for blood testing. Reasons for non-participation in the survey included unsuccessful attempts to reach a participant at home and lack of time to participate in the study. Reasons for not agreeing to blood testing included concerns about confidentiality, the invasiveness of the request and religious reasons.

The final study sample comprised 2,220 participants (705 men and 1,515 women) for the survey and body measures and 1,005 participants (306 men and 699 women) for blood testing after excluding 79 survey respondents with missing responses on questions about socioeconomic factors or the depression screening instrument. The final study sample underrepresented men with respect to the gender structure of the 2016 DUCS population and both men and women in the youngest age group with respect to the 2016 DUCS population. The study sample overrepresented men in higher age groups and women in middle age groups (Supplementary Table S1 and Supplementary Figure S1).

### *Data collection*

Field workers visited study participants' homes to conduct computer-assisted personal interviews, take body measures and conduct blood tests. The survey included adapted versions of pre-existing screening instruments for angina pectoris, depression, alcohol dependence, cognitive impairment and food insecurity. Height, weight and blood pressure were measured. Point-of-care blood testing used the CareSens Blood Glucose Monitoring System and the HemoCue Hemoglobin 201+ Analyzer.

### *Description of variables*

#### *Depressive symptoms*

Depressive symptoms were assessed using a short version of the Center for Epidemiological Studies-Depression Scale with ten items (CES-D-10) (Andresen et al., 1994). A CES-D-10 score ≥10 is indicative of depressive symptoms. The ten items used for the assessment of depression asked about the frequency of eight

negative feelings and two positive feelings in the past week. The items about negative feelings explored if someone was bothered by things that do not usually bother him/her, had trouble keeping attention on what s/he was doing, felt depressed, felt everything was an effort, felt fearful, slept restless, felt lonely or could not get going. The items about positive feelings explored if the respondent felt hopeful about the future or had been happy. Response options included rarely or none of the time (<1 day; 0 points), some or a little of the time (1–2 days; 1 point), occasionally or a moderate amount of time (2–4 days; 2 points) and all the time (5–7 days; 3 points). The scoring was reversed for positive symptoms.

### Age, chronic conditions and health status

Age was categorized into 5-year age groups for 40 to 69 years and one age group for over 70 years. Overall, 13 chronic conditions were assessed. Chronic cough, kidney disease, HIV, hypercholesterolemia, stroke and tuberculosis were based on self-reporting of either a prior diagnosis or ever being treated for a condition. Anemia, diabetes, hypertension, obesity and underweight were defined based on thresholds for hemoglobin, blood glucose, blood pressure and the body mass index, respectively. The hemoglobin threshold was adjusted for smoking and African origin (Sullivan et al., 2008). Ischemic heart disease included a previous diagnosis of heart disease or heart failure, or reporting symptoms of angina pectoris. Angina pectoris symptoms were assessed using a modified Rose Angina Questionnaire (Rose et al., 1977). Signs of cognitive problems were assessed using self-rated memory-and-recall tests, which were adapted from the US Health and Retirement Study (Ofstedal et al., 2005). Study participants who were affected by two or more of these chronic conditions at the time of the study were considered multimorbid. Health status was assessed in terms of self-reported health parameters and any limitation in six activities of daily living. Self-reported health was assessed on a five-item Likert scale ranging from very good to very bad. The six assessed activities of daily living were walking across a room, dressing, bathing, eating, getting into or out of the sleeping place and using the toilet (Supplementary Table S2).

### Control variables

Control variables included substance use and socioeconomic factors. Substance use included smoking and alcohol drinking. Smoking was self-reported current or ever smoking. CAGE questions were asked to assess problem drinking and potential alcohol problems. An affirmative answer to two or more CAGE questions was interpreted as signs of alcohol problems. Social variables included study participants' sex, religion, country of origin, marital status, number of children, literacy level, years of formal education and work status. Economic variables included food insecurity in the household and household wealth. Food insecurity was assessed based on the reported unavailability of food in the house during the last year. Response options were never, rarely (once or twice), sometimes (3–10 times) and often (>10 times). Household wealth was assessed through a principal component analysis of the availability and amount of 30 assets in the household. The first component was chosen and normalized (0–100) for the wealth index. Location-fixed effects were included for the seven main streets around which the sampled households were located.

### Data analysis

We described the characteristics of study participants using the median and interquartile range (IQR) for continuous variables

and frequencies for categorical variables. Statistical differences between groups of study participants were assessed using the Wilcoxon rank-sum test for continuous variables and the Pearson $\chi^2$ test for categorical variables. To estimate the prevalence of depressive symptoms and chronic conditions, we used univariable linear regressions. We estimated the prevalence of depressive symptoms overall, by age and sex and conditional on having chronic conditions.

To examine associations between depressive symptoms and predictor variables, we used univariable and multivariable logistic regressions. First we estimated univariable regressions in which age, chronic conditions and health status variables were individually included as independent variables. Second we estimated a multivariable regression in which age, chronic conditions and health status variables were jointly included as independent variables without control variables. Third we estimated a multivariable regression in which age, chronic conditions and health status variables were jointly included as independent variables while adjusting for substance use and socioeconomic factors.

Standard errors were estimated using the Huber–White sandwich estimator. All estimations used 100 multiple imputations by chained equations for missing data and weights to reproduce the age and sex structure of the DUCS population in 2019–20, which corresponds to the latest available data from this health and demographic surveillance system. The level of statistical significance was set at $P < 0.05$. All analyses were conducted in Stata SE 15.1.

### Ethical considerations

Ethical approval was received from the institutional review boards of Muhimbili University of Health and Allied Sciences, Tanzania (2015-04-22/AEC/Vol.IX/82) and Harvard T.H. Chan School of Public Health, USA (14-4282). Participants gave written informed consent before participation and, where applicable, again before blood collection and testing. Participants who were unable to write could use an ink fingerprint as a signature.

## Results

### Sample characteristics

Of the 2,220 study participants aged over 40 years, 31.8% were men and 68.2% were women. The median age was 50 years (IQR 44–59). Almost all (99.0%) participants were born in Tanzania. Islam was the religion for 54.1% and Christianity for 45.9%. Most study participants were married or cohabitating (70.9%) and had attended 0–6 years of schooling (77.7%). Several study participants were homemakers (34.2%) or working (46.9%). Those not working (18.9%) had retired, were sick or on leave or unable to work due to a disability. Almost half (48.5%) had at least once during the past year no food in the house. One in eight (12.6%) study participants experienced food insecurity >10 times during the past year. Present smoking was reported by 4.3% and ever smoking by 13.3%. A share of 7.8% showed signs of alcohol problems. Age, marital status, literacy, work status, food insecurity and household location showed an association with depressive symptoms (Table 1 and Supplementary Table S3).

Most (89.1%) of the participants had one or more chronic conditions and 61.1% were multimorbid with over two chronic conditions. The most prevalent chronic conditions were hypertension (51.1%), anemia (34.1%), obesity (32.4%), diabetes (31.4%)

**Table 1.** Sociodemographic characteristics and substance use of study participants

| | Total | CES-D-10 < 10 | CES-D-10 ≥ 10 | |
|---|---|---|---|---|
| | $N \leq 2{,}220$ | $N \leq 1{,}520$ | $N \leq 700$ | P |
| Socioeconomic factors | | | | |
| Age (years), n = 2,220 | 50 (44–59) | 50 (44–59) | 51 (45–60) | 0.001 |
| Sex, n = 2,220 | | | | 0.19 |
| Male | 705 (31.8) | 496 (32.6) | 209 (29.9) | |
| Female | 1,515 (68.2) | 1,024 (67.4) | 491 (70.1) | |
| Country of origin, n = 2,220 | | | | 0.91 |
| Tanzania | 2,197 (99.0) | 1,504 (98.9) | 693 (99.0) | |
| Other | 23 (1.0) | 16 (1.1) | 7 (1.0) | |
| Religion, n = 2,220 | | | | 0.46 |
| Islam | 1,202 (54.1) | 815 (53.6) | 387 (55.3) | |
| Christianity | 1,018 (45.9) | 705 (46.4) | 313 (44.7) | |
| Marital status, n = 2,220 | | | | <0.001 |
| Married or cohabitant | 1,575 (70.9) | 1,118 (73.6) | 457 (65.3) | |
| Widowed | 381 (17.2) | 227 (14.9) | 154 (22.0) | |
| Never married or separated | 264 (11.9) | 175 (11.5) | 89 (12.7) | |
| Number of children, n = 2,220 | | | | 0.099 |
| 0 | 63 (2.8) | 44 (2.9) | 19 (2.7) | |
| 1–2 | 499 (22.5) | 322 (21.2) | 177 (25.3) | |
| ≥3 | 1,658 (74.7) | 1,154 (75.9) | 504 (72.0) | |
| Can read and/or write, n = 2,220 | 1,864 (84.0) | 1,309 (86.1) | 555 (79.3) | <0.001 |
| Formal education, n = 2,220 | | | | 0.14 |
| 0–6 school years | 1,725 (77.7) | 1,176 (77.4) | 549 (78.4) | |
| 7–10 school years | 108 (4.9) | 67 (4.4) | 41 (5.9) | |
| ≥10 school years | 387 (17.4) | 277 (18.2) | 110 (15.7) | |
| Work status, n = 2,220 | | | | <0.001 |
| Homemaker | 759 (34.2) | 527 (34.7) | 232 (33.1) | |
| Working | 1,041 (46.9) | 754 (49.6) | 287 (41.0) | |
| Not working | 420 (18.9) | 239 (15.7) | 181 (25.9) | |
| No food in house, n = 2,220 | | | | <0.001 |
| Never | 1,143 (51.5) | 849 (55.9) | 294 (42.0) | |
| Rarely (once or twice) | 626 (28.2) | 468 (30.8) | 158 (22.6) | |
| Sometimes (3–10 times) | 171 (7.7) | 90 (5.9) | 81 (11.6) | |
| Often (more than 10 times) | 280 (12.6) | 113 (7.4) | 167 (23.9) | |
| Household wealth index (0–100), n = 2,177 | 18 (15–21) | 18 (15–21) | 17 (15–21) | 0.084 |
| Ward, N = 2,220 | | | | 0.008 |
| Ukonga (with 4 major streets) | 1,224 (55.1) | 809 (53.2) | 415 (59.3) | |
| Gongolamboto (with 3 major streets) | 996 (44.9) | 711 (46.8) | 285 (40.7) | |
| Substance use | | | | |
| Currently smoking, n = 2,218 | 96 (4.3) | 65 (4.3) | 31 (4.4) | 0.87 |
| Ever smoked, n = 2,218 | 296 (13.3) | 193 (12.7) | 103 (14.7) | 0.19 |
| Signs of alcohol problems, n = 2,218 | 172 (7.8) | 118 (7.8) | 54 (7.7) | 0.97 |

*n* (%) or median (IQR).

and ischemic heart disease (11.9%). All other conditions affected <10% of the participants.

Anemia, ischemic heart disease, signs of cognitive problems, ever diagnosed with stroke or tuberculosis, multimorbidity, moderate or bad health on the day of survey and limitations in activities of daily living were associated were more common among the study participants with depressive symptoms (Table 2 and Supplementary Table S4).

### Prevalence of depressive symptoms

The estimated prevalence of depressive symptoms was 30.7% (95% CI 28.5–32.9) after weighing the study sample to match the age and sex structure of the population living in the Ukonga and Gongolamboto wards in 2019–20. Depressive symptoms peaked for those aged over 70 years at 48.0% (95% CI 40.4–55.7). A second, local peak was estimated among 45–49-year-olds at 33.6% (95% CI 28.9–38.4). Across the age groups, the highest prevalence of depressive symptoms was estimated for people with ischemic heart disease (59.6% [95% CI 53.1–66.1]), followed by those ever diagnosed with tuberculosis (48.6% [95% CI 37.3–

60.0]), those with signs of cognitive problems (46.2% [95% CI 37.3–55.2]) and those ever diagnosed with stroke (44.8% [95% CI 34.4–55.2]). The prevalence of depressive symptoms among middle-aged and elderly not affected by any of the assessed chronic conditions was 23.0% (95% CI 12.8–33.3). It was 34.1% (95% CI 29.6–38.5) among those who were multimorbid. Among those who rated their current health as bad or very bad, the estimated prevalence of depressive symptoms was 43.0% (95% CI 35.5–50.5). Among those rating their current health as good or very good, the prevalence was 23.3% (95% CI 20.1–26.6). For ≥40-year-olds with limitations in activities of daily living, the estimated prevalence was 40.2% (95% CI 34.5–45.8) (Figures 1, 2 and Supplementary Table S5).

### Associations of depressive symptoms with age, chronic conditions and health status

Being 45–49 years old (OR 1.35 [95% CI 1.04–1.75]) or ≥70 years old (OR 2.35 [95% CI 1.66–3.33]) increased the odds of depressive symptoms compared to being 40–44 years old in a univariable regression (column 1 of Table 3). Adjusting for chronic conditions

**Table 2.** Depressive symptoms, chronic conditions and health status of study participants

| | Total | CES-D-10 < 10 | CES-D-10 ≥ 10 | |
|---|---|---|---|---|
| | N ≤ 2,220 | N ≤ 1,520 | N ≤ 700 | P |
| **Depressive symptoms** | | | | |
| CES-D-10 score, n = 2,220 | 7 (5–11) | 6 (4–7) | 14 (11–17) | <0.001 |
| **Chronic condition** | | | | |
| Hypertension, n = 2,168 | 1,108 (51.1) | 761 (51.1) | 347 (51.0) | 0.96 |
| Anemia, n = 977 | 333 (34.1) | 213 (31.8) | 120 (39.1) | 0.026 |
| Obesity, n = 2,126 | 688 (32.4) | 457 (31.2) | 231 (34.9) | 0.093 |
| Diabetes, n = 978 | 307 (31.4) | 212 (31.5) | 95 (31.0) | 0.88 |
| Ischemic heart disease, n = 2,219 | 264 (11.9) | 113 (7.4) | 151 (21.6) | <0.001 |
| Signs of cognitive problems, n = 2,213 | 145 (6.6) | 79 (5.2) | 66 (9.5) | <0.001 |
| HIV, n = 2,209 | 112 (5.1) | 70 (4.6) | 42 (6.1) | 0.15 |
| High cholesterol, n = 2,216 | 110 (5.0) | 81 (5.3) | 29 (4.2) | 0.24 |
| Stroke, n = 2,218 | 107 (4.8) | 63 (4.1) | 44 (6.3) | 0.028 |
| Tuberculosis, n = 2,216 | 107 (4.8) | 52 (3.4) | 55 (7.9) | <0.001 |
| Underweight, n = 2,126 | 89 (4.2) | 60 (4.1) | 29 (4.4) | 0.76 |
| Chronic cough, n = 2,215 | 73 (3.3) | 46 (3.0) | 27 (3.9) | 0.30 |
| Kidney disease, n = 2,216 | 64 (2.9) | 42 (2.8) | 22 (3.2) | 0.61 |
| **Multimorbidity** | | | | |
| No chronic conditions, n = 941 | 103 (10.9) | 81 (12.4) | 22 (7.6) | 0.029 |
| ≥2 chronic conditions, n = 941 | 575 (61.1) | 378 (58.0) | 197 (68.2) | 0.003 |
| **Health status** | | | | |
| Health today, n = 2,218 | | | | <0.001 |
| Good or very good | 840 (37.9) | 633 (41.7) | 207 (29.6) | |
| Moderate | 1,161 (52.3) | 756 (49.8) | 405 (57.9) | |
| Bad or very bad | 217 (9.8) | 129 (8.5) | 88 (12.6) | |
| Limitations in activities of daily living, n = 2,219 | 373 (16.8) | 235 (15.5) | 138 (19.7) | 0.012 |

n (%) or median (IQR).

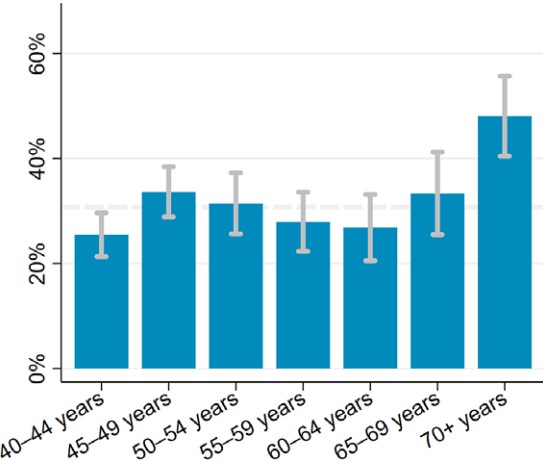

**Figure 1.** Prevalence of depressive symptoms among ≥40-year-olds in the Ukonga and Gongolamboto wards of Dar es Salaam by age. Dashed line indicates average prevalence of depressive symptoms. Estimates weighted to match age and sex structure in the Dar es Salaam Urban Cohort Study 2019–20.

3 of Table 3). Associations between depressive symptoms and HIV, diabetes, kidney disease, chronic cough or underweight were not significant. Multimorbidity was positively associated with depressive symptoms in a univariable regression (OR 1.53 [95% CI 1.23–1.90]) (column 1 of Table 3). The association between depressive symptoms and multimorbidity ceased in multivariable regressions that adjusted for age and individual chronic conditions or for age, chronic conditions and control variables (columns 2 and 3 of Table 3).

Older adults who reported good or very good health had lower odds of depressive symptoms in both univariable regression (OR 0.48 [95% CI 0.35–0.66]) and multivariable regression after adjusting for age and chronic conditions (columns 1 and 2 of Table 3). The association between depressive symptoms and good or very good health ceased to be significant in a multivariable regression that also adjusted for substance use and socioeconomic factors (column 3 of Table 3). Adults with limitations in activities of daily living had higher odds of depressive symptoms (OR 1.35 [95% CI 1.07–1.70]) only in a univariable regression (column 1 of Table 3).

reduced the odds of depressive symptoms for ≥70-year-olds (column 2 of Table 3). The association between depressive symptoms and age over 70 years ceased in multivariable regressions that adjusted for chronic conditions, substance use and socioeconomic factors (column 3 of Table 3).

In univariable regressions, ischemic heart disease (OR 3.43 [95% CI 2.64–4.46]), tuberculosis (OR 2.42 [95% CI 1.64–3.57]), signs of cognitive problems (OR 1.90 [95% CI 1.35–2.67]), stroke (OR 1.56 [95% CI 1.05–2.32]) and anemia (OR 1.32 [95% CI 1.01–1.71]) were associated with higher odds of depressive symptoms (column 1 of Table 3). In multivariable regression models, ischemic heart disease and tuberculosis remained associated with depressive symptoms, but the odds decreased. Stroke, signs of cognitive problems and anemia ceased to be associated with depressive symptoms when adjusting for age and other chronic conditions. A diagnosis of high cholesterol was associated with lower odds of depressive symptoms after adjusting for age and chronic conditions or for age, other chronic conditions and control variables (columns 2 and

## Discussion

### Summary of findings

Using the ten-item version of the CES-D-10, we estimated a prevalence of depressive symptoms of 30.7% among the ≥40-year-olds living in the Ukonga and Gongolamboto wards in Dar es Salaam. In univariable regressions, depressive symptoms were associated with age, ischemic heart disease, tuberculosis, signs of cognitive problems, anemia, multimorbidity, self-reported health and limitations in activities of daily living. Adjusting for age and other chronic conditions or for age, other chronic conditions, substance use and socioeconomic factors tended to reduce the associations of individual chronic conditions with depressive symptoms. Having ever been diagnosed with ischemic heart disease and tuberculosis remained independent predictors of depressive symptoms in multivariable regressions. We found no evidence for associations between depressive symptoms and obesity, diabetes, HIV, kidney disease, chronic cough and underweight.

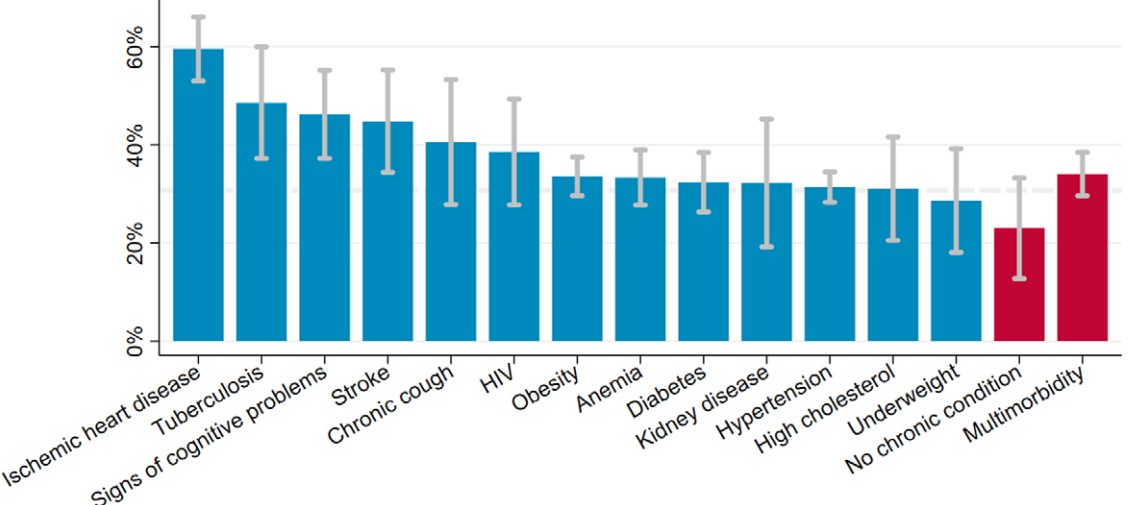

**Figure 2.** Prevalence of depressive symptoms among ≥40-year-olds in the Ukonga and Gongolamboto wards of Dar es Salaam by chronic condition and multimorbidity. Dashed line indicates average prevalence of depressive symptoms. Estimates weighted to match age and sex structure in the Dar es Salaam Urban Cohort Study 2019–20.

**Table 3.** Association between depressive symptoms and age, chronic conditions and health status among ≥40-year-olds in the Ukonga and Gongolamboto wards of Dar es Salaam

| Depressive symptoms (CES-D-10 ≥ 10), n = 2,220 | Univariable regressions | Multivariable regression | Multivariable regression with control variables |
|---|---|---|---|
| Age | | | |
| Age group (40–44 years) | 1 | 1 | 1 |
| 45–49 years | 1.35 (1.04–1.75)* | 1.37 (1.04–1.8)* | 1.35 (1.01–1.81)* |
| 50–54 years | 1.29 (0.97–1.73) | 1.23 (0.90–1.68) | 1.13 (0.80–1.59) |
| 55–59 years | 1.12 (0.81–1.53) | 1.04 (0.74–1.45) | 0.94 (0.65–1.36) |
| 60–64 years | 1.05 (0.74–1.49) | 0.97 (0.67–1.41) | 0.83 (0.55–1.26) |
| 65–69 years | 1.34 (0.91–1.97) | 1.18 (0.76–1.83) | 0.88 (0.53–1.46) |
| ≥70 years | 2.35 (1.66–3.33)*** | 1.88 (1.26–2.80)** | 1.29 (0.81–2.06) |
| Chronic condition | | | |
| Ischemic heart disease | 3.43 (2.64–4.46)*** | 3.01 (2.26–4.01)*** | 2.73 (2.01–3.71)*** |
| Tuberculosis | 2.42 (1.64–3.57)*** | 2.12 (1.36–3.3)*** | 2.02 (1.24–3.27)** |
| Signs of cognitive problems | 1.90 (1.35–2.67)*** | 1.37 (0.93–2.02) | 1.30 (0.86–1.95) |
| Stroke | 1.56 (1.05–2.32)* | 1.16 (0.76–1.79) | 1.15 (0.71–1.86) |
| Chronic cough | 1.29 (0.79–2.09) | 1.05 (0.63–1.76) | 1.13 (0.63–2.01) |
| HIV | 1.31 (0.88–1.94) | 1.18 (0.77–1.80) | 0.94 (0.57–1.55) |
| Obesity | 1.16 (0.96–1.41) | 1.25 (0.97–1.60) | 1.28 (0.97–1.69) |
| Anemia | 1.32 (1.01–1.71)* | 1.18 (0.86–1.62) | 1.13 (0.80–1.61) |
| Diabetes | 1.07 (0.82–1.40) | 1.02 (0.75–1.40) | 0.99 (0.70–1.39) |
| Kidney disease | 1.15 (0.68–1.94) | 0.95 (0.54–1.68) | 1.25 (0.69–2.26) |
| Hypertension | 1.00 (0.83–1.19) | 0.91 (0.72–1.15) | 0.81 (0.63–1.04) |
| High cholesterol | 0.78 (0.50–1.20) | 0.57 (0.34–0.94)* | 0.56 (0.34–0.94)* |
| Underweight | 1.19 (0.77–1.85) | 1.05 (0.67–1.67) | 0.93 (0.55–1.57) |
| Multimorbidity | 1.53 (1.23–1.90)*** | 0.99 (0.68–1.43) | 1.04 (0.70–1.55) |
| Health status | | | |
| Health today (bad or very bad) | 1 | 1 | 1 |
| Moderate | 0.79 (0.58–1.06) | 0.95 (0.68–1.31) | 0.95 (0.67–1.34) |
| Good or very good | 0.48 (0.35–0.66)*** | 0.66 (0.46–0.94)* | 0.71 (0.48–1.05) |
| Limitations in activities of daily living | 1.35 (1.07–1.70)* | 1.04 (0.80–1.36) | 0.90 (0.67–1.21) |
| Control variables | | | |
| Substance use | No | No | Yes |
| Socioeconomic factors | No | No | Yes |
| Constant | Yes | 0.35 (0.23–0.53)*** | 0.64 (0.26–1.57) |

OR (95% CI). Estimates weighted to match age and sex structure in the Dar es Salaam Urban Cohort Study 2019–20. Regression estimates of control variables are provided in Supplementary Table S6.
***$P < 0.001$
**$P < 0.01$
*$P < 0.05$.

### Depressive symptoms and age

We estimated depressive symptoms among ≥70-year-olds to be the highest (48.0%). Previous cross-sectional studies assessed depression and depressive symptoms in rural Hai and Moshi districts of the Kilimanjaro region in Tanzania, respectively. Based on the diagnostic criteria of the fourth edition of the *Diagnostic and Statistical Manual of Mental Disorders*, 21.2% of the people aged over 70 years in the Hai district were diagnosed with depression (Mlaki et al., 2021). Based on the 15-item Geriatric Depression Scale, 44.4% of the people aged over 60 years in the Moshi district had depressive symptoms (Adams et al., 2020). Our estimates for the prevalence of depressive symptoms in ≥60-year-olds ranged from 26.8% to 48.0%. Differences in these estimates could, among other factors, result from the use of different screening tools, screening tools versus clinical assessment, and wealth differences between Dar es Salaam and rural regions of Tanzania.

### Depressive symptoms and ischemic heart disease

We estimated that 59.6% of ≥40-year-olds with ischemic heart disease had depressive symptoms. The co-occurrence of depression

and heart disease has been described before (Vaccarino et al., 2019). A meta-analysis of prospective studies has found that depression increased the risk for subsequent coronary heart disease and myocardial infarction by 30% (Gan et al., 2014). Depression in people with heart disease has been described as a risk factor for adverse medical outcomes due to lower treatment adherence and higher mortality and increased healthcare costs (Lichtman et al., 2014; Vaccarino et al., 2019).

### Depressive symptoms and tuberculosis

Among ≥40-year-olds with tuberculosis, we estimated a prevalence of depressive symptoms of 48.6%. Another study among people with tuberculosis attending clinics in Temeke municipality of Dar es Salaam used the Patient Health Questionnaire-9 to assess the prevalence and severity of depression. Among the 46.9% of people suffering from tuberculosis coupled with depression, 33.6% had mild, 13.3% moderate and 0% moderately to severe, or severe depression (Buberwa, 2013). Depression has also been found to lower treatment adherence and increase mortality in people with tuberculosis (Sweetland et al., 2014, 2017).

### Depressive symptoms and cognitive impairment

We estimated that 46.2% of ≥40-year-olds with signs of cognitive problems showed depressive symptoms. We observed an association of depressive symptoms with signs of cognitive problems in a univariable regression. This association was explained by age, other chronic conditions and health status in multivariable regressions. An association between cognitive impairment and depression has been previously described. In another cross-sectional study among 60-year-olds living in rural northern Tanzania, the prevalence of depressive symptoms was higher among those with a self-reported history of cognitive impairment (Adams et al., 2020). A systematic review has identified cognitive impairment as a predictor of depression among elderly Caucasians (Djernes, 2006).

### Depressive symptoms and stroke

We estimated the prevalence of depressive symptoms of 44.8% among ≥40-year-olds with a previous stroke. The association between depressive symptoms and ever been diagnosed with a stroke ceased to be significant in multivariable regressions that adjusted for age, other chronic conditions and health status. Post-stroke depression is a known phenomenon . Physical disability, cognitive impairment, low education and divorced marital status, among other factors, have been associated with post-stroke depression in sub-Saharan Africa (Ojagbemi et al., 2017).

### Depressive symptoms and anemia

Anemia may result from various causes, including being a woman of reproductive age, nutritional deficiencies, hematologic diseases and cancers. The estimated prevalence of depressive symptoms among ≥40-year-olds with anemia was 33.4%. Anemia was associated with higher odds of depressive symptoms only in a univariable regression. A meta-analysis of observational studies has reported a positive association between anemia and depression even after adjusting for comorbidities, alcohol use, smoking and education. The meta-analysis also found evidence of a publication bias (Lee and Kim, 2020).

### Depressive symptoms and cholesterol

Adults aged over 40 years with self-reported high cholesterol had a prevalence of depressive symptoms of 31.1%. High cholesterol was associated with lower odds of depressive symptoms in multivariable regressions which adjusted for the presence of age, other chronic conditions and health status. Lower odds of depression in the presence of lower low-density lipoprotein blood serum levels have been described in a meta-analysis (Persons and Fiedorowicz, 2016). Another meta-analysis has reported a negative association between depression and total cholesterol and a negative but insignificant association between depression and low- and high-density lipoprotein serum levels (Shin et al., 2008).

### Depressive symptoms and multimorbidity

We estimated the prevalence of depressive symptoms of 34.1% among multimorbid ≥40-year-olds. In comparison, we estimated the prevalence of depressive symptoms of 23.0% for a group not affected by any of the assessed chronic conditions. Multimorbidity increased the odds of depressive symptoms in a univariable regression. It ceased to be a predictor of depressive symptoms when adjusting for age, individual underlying chronic conditions and health status. An increased risk of depression among adults with multimorbidity compared to those without has been reported in a meta-analysis (Read et al., 2017). Our findings suggest that multimorbidity may not add to the risk of depressive symptoms on its own.

### Depressive symptoms and health status

The prevalence of depressive symptoms was 23.3% among ≥40-year-olds who rated their health as good or very good as compared to 43.0% among those who rated their health as bad or very bad. The prevalence of depressive symptoms was 40.2% among those with limitations in activities of daily living. Good self-rated health was negatively associated and any limitation in activities of daily living was positively associated with depressive symptoms in univariable regressions. Limitations in activities of daily living ceased to be associated with depressive symptoms after adjusting for age, chronic conditions and health status. Good self-rated health ceased to be associated with lower odds of depressive symptoms after adjusting also for substance use and socioeconomic factors. A higher risk for depression among elderly people with poor self-rated health has been previously reported by a meta-analysis (Chang-Quan et al., 2009). A Chinese study of middle-aged and elderly has also found mixed evidence on the association of depressive symptoms with limitations in activities of daily living, depending on the severity of the limitations (He et al., 2019).

### Practical implications

A gap in diagnosis and treatment of mental health problems has been described for Tanzania. Insufficient training for healthcare providers, shortage of human resources and psychiatric wards as well as lack of diagnostic and treatment methods have been named as barriers for adequate mental health care in Dar es Salaam (Ambikile and Iseselo, 2017). Information about the high prevalence of depressive symptoms might reduce stigmatizing attitudes and lead to higher prioritization of mental health care. Furthermore, detecting and treating depression will benefit the treatment

outcomes of chronic diseases as depression can lower adherence to medication or other forms of treatment.

First, this study indicates that depressive symptoms are common among middle-aged and elderly people in the Dar es Salaam region. The prevalence of depressive symptoms was highest among the oldest. We observed a second, lower peak of the prevalence of depressive symptoms among 45–49-year-olds. Second, this study examined the association of chronic conditions with depressive symptoms. Awareness of the predictors of depressive symptoms can help target depression screening, for instance, within specific health services. Third, this study found that chronic conditions and socioeconomic factors affected the extent to which depressive symptoms occurred. In sum, these findings suggests that a combination of social, economic and healthcare interventions could reduce a substantial risk of depression among the older population in peri-urban Dar es Salaam.

### Study strengths and limitations

Strengths of this study include, first, that data were collected from a general population. Second, we used a depression screening instrument that has been previously validated (Andresen et al., 1994; Boey, 1999; Irwin et al., 1999) and used in studies in sub-Saharan Africa (Tomita and Burns, 2013; Kilburn et al., 2018; Geldsetzer et al., 2019). Third, we studied a wide range of chronic conditions and their associations with depressive symptoms. Limitations of this study include, first, that the CES-D-10 has only been validated in younger populations in Tanzania (Kilburn et al., 2018). Second, also the diagnosis of several conditions was based on screening instruments not validated in the study population. Third, some chronic conditions were self-reported, which might have led to a bias in their estimated prevalence. Finally, our study sample was affected by selection bias and its cross-sectional design prevents making causal inferences.

### Conclusion

Depressive symptoms were estimated to affect one in three ≥40-year-old people in the peri-urban Ukonga and Gongolamboto wards of Dar es Salaam. Those with compromised health – notably the oldest among others – had a higher risk of depressive symptoms. Targeted screening among the oldest people and those affected by ischemic heart disease, tuberculosis, cognitive problems or multimorbidity might help detect those at highest risk for depression.

**Open peer review.** To view the open peer review materials for this article, please visit http://doi.org/10.1017/gmh.2023.17.

**Supplementary material.** The supplementary material for this article can be found at http://doi.org/10.1017/gmh.2023.17.

**Data availability statement.** Data underlying this study are available on a reasonable request from J.K.R.

**Acknowledgements.** L.-M.S. acknowledges the support of Else Kröner-Fresenius-Stiftung within the Heidelberg Graduate School of Global Health.

**Author contribution.** L.-M.S. and S.K. conceived the study, conducted data analysis and wrote the manuscript. T.B., J.K. and J.K.R. acquired funding for data collection. G.H.L., P.K. and J.K. led study implementation activities in Dar es Salaam. All authors contributed to revisions and reviewed the manuscript.

**Financial support.** Research reported in this publication was supported by the National Institute on Aging of the National Institutes of Health under award number P30AG024409.

**Competing interest.** The authors declare that they have no competing interests.

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
