## [Reviewer Report]

Dear Professor Bass, dear Dixon,

dear Jess and Laeti,

we are delighted to send you the invited submission of our study “Depressive symptoms and their associations with age, chronic conditions, and health status: A community-based study among middle-aged and elderly people in peri-urban Tanzania”.

Depression is known as one of the most common mental health conditions and an increasing challenge to public health in Sub-Saharan Africa. In Tanzania, rapid urbanization, population ageing and changing patterns of chronic infectious and non-infectious diseases pose risk factors for depression. Especially older adults can be at high risk for depression in the context of other chronic conditions. Previous studies have found high prevalence of depression in rural areas of Tanzania to which provision of health care is not yet adapted. To our knowledge, this is the first community-based study to investigate depression among middle-aged and elderly in an urban area in Tanzania. 

The strengths and unique contributions of the study are: 

1. Presentation of prevalence estimates for depressive symptoms based on a large community-based sample.

2. Assessment of a wide range of chronic conditions, based on which we identified subgroups that have higher odds for depression. 

3. Inclusion of variables on the general health status including self-rated health and limitations in activities of daily living.

My co-authors and I used the opportunity to submit of Global Mental Health to submit a longer research article for an in-depth presentation and discussion of our findings. Would be delighted if the manuscript could fit for the journal as it is, and we are also happy to incorporate changes suggested by the editorial team or reviewers.

Please contact me with questions anytime.

Best wishes,

Stefan Kohler

(on behalf of all authors)

---

## [Reviewer Report]

*Comments to Author*: In this study the authors assess the prevalence of depressive symptoms and their association with age, chronic conditions, and health status in 2,220 adults aged 40 years and over in 2 wards of Dar es Salaam, Tanzania using the 10-item version of the CES-D-10 scale to assess depression. The prevalence of depressive symptoms was 30.7% and these symptoms were associated with ischaemic heart disease, tuberculosis, signs of cognitive problems, stroke, anaemia and limitation of activities of daily living. They were also higher in the 70 years and over age group. 

This is the first community-based study of the prevalence of depression in an urban area in Tanzania, and the first study to look at the associations with chronic conditions, multimorbidity, self related health and limitation in activities of daily living. This is an important area of research and the findings are interesting and helpful. 

There are, however, limitations, most of which have been acknowledged by the authors as listed in points 1 – 5 below with additional comments from me.

1. This is a cross sectional study which doesn’t allow for causal influences.

2. The diagnosis of some of the chronic conditions was based on self-report which can lead to over – or under estimation of prevalence estimates. This is a major limitation – please see below. 

3. The diagnosis of several other conditions, including depressive symptoms, was based on screening instruments. The CES-D-10 is a frequently used tool but, as the authors point out, it has not been validated in Swahili or in the elderly in Tanzania. It would be worth the authors commenting on whether it has been validated in other countries in sub-Saharan Africa, or even in other low and middle income countries (LMICs). As the authors point out it is only a screening, not a diagnostic, tool and therefore there is no estimate of the severity of the depression, and the need for psychiatric treatment. It would be helpful if the authors reported how many people had previously been diagnosed with depression, and what proportion received a pharmaceutical or non-pharmaceutical intervention for this. How many have ever taken anti-depressants, and how many were on them currently? Screening instruments often tend to give higher prevalence estimates as they are designed to try to identify everyone with the particular problem they are screening for.

4. Chronic conditions such as anxiety, chronic pain or musculoskeletal disorders, which may be associated with depression, were not included. I am surprised that anxiety wasn’t assessed. It is also disappointing that there was no measure of pain or musculoskeletal disorders as these can certainly be associated with depression and may have a high prevalence within this population. 

5. The authors point out that the data are based on an urban area in eastern Africa and so are not necessarily generalisable but do report how their statistical analyses potentially make these more generalisable. 

Further comments and questions:

Methods

The study was conducted from June 2017 – July 2018, so more than 4 years ago.

In relation to the study population it is not stated how the 4,850 people were randomly selected. Of these less than half (2,270) were interviewed at home, and a further 50 were excluded due to missing key data giving a final study sample of 2,220. It would be helpful if there were some demographic data on people who were originally randomly selected so that we can see how representative the 2,220 people are of the 4,850. Also, it would helpful to know how the original cohort were “randomly selected”.

Data collection tools

How reliable is the Rose Angina questionnaire in LMIC settings? Were other data used to justify the prevalence of ischaemic heart disease, e.g. previous hospital admissions, ECGs etc. 

How reliable is the US HRS cognitive test battery in this environment? Are there examples of where it has previously been used in LMICs? 

How were the half of the study’s participants randomly selected to participate in point-of-care blood glucose and haemoglobin testing? 

Is there a reference to justify the common cut off of score >=10 as indicative for depressive symptoms on the CES-D-10 scale? Were other depression scales considered? It might be worth the authors discussing why the CES-D-10 was chosen.

How was age determined in those people who did not have a birth certificate or did not know their date of birth or even year of birth, as is quite common in such settings? 

Kidney disease and hypercholesteremia were based on self-report so these are likely to be major underestimates as it is likely that many people will not have been tested for these previously. 

In those who underwent blood testing was the blood glucose fasting or non-fasting? How was diabetes defined based on this? 

How was blood pressure measured (e.g. after sitting and resting for 5 minutes, and then repeated twice 5 minutes apart, as per WHO recommendations?) What cut off level was used to define high blood pressure?

I am interested that difficulties in one or more activities of daily living assessed counted as “limitations in activities of daily living” as this may potentially overestimate problems but didn’t appear to. The six assessed activities seem appropriate.

In relation to ethical considerations how was consent taken from people who couldn’t write? 

Results 

It would be helpful to know about previous diagnosis rates for diabetes and anaemia and how many new cases were picked up by the screening blood test. 

Also, as mentioned above, it would be helpful to know what further justification there was for ischaemic heart disease apart from the Rose questionnaire. 

Discussion

Discussion is clearly set out relating to the different components of the research and is generally easy to follow and appropriate. 

As the authors point out stigma could lead to under-reporting of conditions such as HIV as these data were dependent on self-report.

As the authors point out clearly just the presence or absence of a condition such as HIV or stroke does not provide much detail as the severity of these conditions, and how they impact on individuals, can be very variable. 

As the authors point out the CES-D-10 assesses symptoms relating to physical functioning that might occur in adults with limitations to activities of daily living, leading to higher CES-D-10 scores without being depressed. 

As regards practical implications I would entirely agree with the authors about the need for further research in this area, and the need for further training for healthcare professionals to recognise depression. It might also be worth the authors commenting further on potential interventions for those diagnosed with depression in this setting.

Overall I think this is an important research study utilising robust methods and reporting interesting and important findings. As the authors recognise, there are several limitations.

---

## [Reviewer Report]

*Comments to Author*: Dear authors,

Thank you for this fascinating and important study – raising the profile of mental health comorbidity in Tanzania, where research on such issues is still in its infancy. This is a valuable contribution to the literature, particularly in focusing on a community, naturalistic sample in a non-rural area, both for health professionals working in Tanzania as well as other Sub-Saharan countries. The article has a clear writing style with a good structure and concise reporting of the findings, which is extremely helpful in order to reach a wide audience and generally raise awareness of depression as a significant issue to include in health services for adults in Tanzania, whether at the planning, preventative or treatment-providing stages.

I would like to highlight some minor issues which you could consider in different sections. In the Introducion, some mention of similar studies conducted elsewhere in sub-saharan African countries would be helpful, to provide a bit more context – I appreciate you do this in the Discussion, but a brief pointer in the Introduction would be appreciated. 

In the Methods section, it would be informative to understand your reasons for using the CES-D-10 and not repeating the use of a previous study’s cited measure i.e. GDS-15? You mention in the discussion that using the CES-D-10 would tend to find a higher prevalence – it would be helpful to have a note linking this to your decision to use an alternative measure.

You mention that your study had a large amount of women in the sample – how does this compare to other studies in Tanzania, and to the general population in the 2 wards? I am just curious whether this could have any bearing on the findings. A clear statement about the impact of gender in your results would be helpful.

There is much to discuss in your Discussion section! You discuss how your findings compare to two other similar studies in Tanzania and it would be interesting to hear thoughts on what may contribute to the differences between your study and those, apart from the assessment measures e.g. could urbanization be a risk factor for Tanzania? This relates to wanting clearer naming of psychological factors that could serve as underlying precipitating or maintaining factors in these groups that you have identified as vulnerable to depression. Likewise, some clear signposting to future research and next steps for understanding more about the nature of the relationship between variables and implications for treatment would be useful.

The issue of how populations differing in ethnic and cultural background manifest symptoms of depression could be included. I believe there is literature discussing whether some groups show more physical symptoms of depression, which might conflate and “over-indicate” depression. This might be particularly true for community samples in Tanzania and explain why the 2 studies mentioned have such differing rates of depression i.e. full assessment yields a more accurate, lower rate.

Lastly, some specific thoughts: on p.9 section 4.8 you begin a sentence “In the following…” – you may want to consider stylistically changing this to something like “In this discussion, we will focus on …” to avoid reader confusion/discomfort. This is a very interesting section and issues are beautifully discussed!

Section 4.11 is also interesting and important – it might be worth linking your reflections that the associations you found with some physical health conditions could affect treatment adherence, as this information is part of the education you would want to disseminate to health professionals.

Thank you again, ahsante sana.

---

## [Reviewer Report]

*Comments to Author*: 1. Please check the manuscript and ensure all grammatical errors and typos are eliminated.

2. Please ensure you have a cutoff for middle age and the elderly. This will make the interpretation of the study findings easier. It will also make the discussion clearer.

3. Tables 1 and 2. Authors do not need the statistical differences by gender, which is not part of the study objective. I suggest they make differences by depression status (yes vs. no). This will lead to the elimination of table 3.

4. The study can do away with the figures. They add no value to the findings. 

5. Table 4: I suggest the authors provide the r square value for models 3 and 4.

6. Table 4: How did the authors have a constant for model 1? Rectify this. 

7. The authors should provide model statistics and model fitness (goodness of fit) in the result section. Based on the current findings, the indication for the various alterations/models is still unclear. 

8. The introduction was based mainly on elderly depression. I request the authors to balance the introduction by introducing both middle age and elderly depression. 

9. The discussion is full of results. Reduce repetition of results and discuss the magnitudes or differences with other studies.

10. For discussion in section 4.1. Please emphasize differences in the tools used. 

11. In the discussion, make comparisons between odds ratios and other measures of relationships other than odds ratios and prevalences or percentages.

---

## [Reviewer Report]

*Comments to Author*: Dear Authors, Thank you for undertaking such a detailed scrutiny of your article and considering how to incorporate reviewers' feedback into your article. I hope with publication you will now receive yet more interesting opinions from a wider audience.

Best wishes!